# On the Reciprocal Relationship between Quantitative and Qualitative Job Insecurity and Outcomes. Testing a Cross-Lagged Longitudinal Mediation Model

**DOI:** 10.3390/ijerph18126392

**Published:** 2021-06-12

**Authors:** Sonia Nawrocka, Hans De Witte, Margherita Brondino, Margherita Pasini

**Affiliations:** 1Research Group Work, Organizational and Personnel Psychology, FPPW, KU Leuven, Oude Markt 13, 3000 Leuven, Belgium; hans.dewitte@kuleuven.be; 2Department of Philosophy, Pedagogy and Psychology, University of Verona, Lungadige Porta Vittoria, 17, 37129 Verona, Italy; margherita.brondino@univr.it (M.B.); margherita.pasini@univr.it (M.P.); 3Optentia Research Unit, North-West University, Vanderbijlpark 1900, South Africa

**Keywords:** quantitative job insecurity, qualitative job insecurity, cross-lagged panel model, conservation of resources theory, burnout, work attitudes, job performance

## Abstract

Prior cross-sectional research indicates that the negative effects of quantitative job insecurity (i.e., threat to job loss) on employees’ wellbeing are fully mediated by qualitative job insecurity (i.e., threat to job characteristics). In the current longitudinal study, we replicated and further extended this view to include a direct effect of qualitative job insecurity on quantitative job insecurity. We explored these reciprocal relations in the context of their concurrent effects on work related outcomes by means of dual-mediation modelling. We identified a wide range of the outcomes, classified as: job strains (i.e., exhaustion, emotional and cognitive impairment), psychological coping reactions (i.e., job satisfaction, work engagement, turnover intention), and behavioral coping reactions (i.e., in-role and extra role performance, counterproductive behavior). We employed a three-wave panel design and surveyed 2003 Flemish employees. The results showed that the dual-mediation model had the best fit to the data. However, whereas qualitative job insecurity predicted an increase in quantitative job insecurity and the outcome variables six months later, quantitative job insecurity did not affect qualitative job insecurity or the outcomes over time. The study demonstrates the importance of qualitative job insecurity not only as a severe work stressor but also as an antecedent of quantitative job insecurity. Herewith, we stress the need for further research on the causal relations between both dimensions of job insecurity.

## 1. Introduction

The literature on organizational change links the volatility of the labor market with ongoing economical, societal, and technological changes [1]. In addition, constant demands to adapt to the dynamic and competitive global markets require organizations to implement a wide range of restructuring strategies. These changes negatively impact employees’ work and the context in which their job is performed [2,3,4]. Consequently, employees might experience an elevated threat over their work-related future [5]. This has sparked an interest, among both scholars and practitioners, in exploring the nature and consequences of employees’ perceived threat of losing a job, defined as quantitative job insecurity [6]. To date, an overwhelming amount of evidence has identified quantitative job insecurity as a severe work stressor with a detrimental effect to employees’ wellbeing [7]. In the last decades, as workplace changes became a natural part of organizational life [8], researchers directed their attention toward less studied qualitative aspect of job insecurity, defined as a perceived threat of loss or negative change to valued job characteristics [9]. A growing number of research stresses the importance of the qualitative dimension of job insecurity as a common work stressor with negative consequences to employees’ health [10], work attitudes [11], and performance [12]. Subsequently, the current literature on job insecurity conceptualizes job insecurity as a two-dimensional concept, with each dimension emphasizing distinct aspects of work-related precariousness [13].

Despite the comprehensive knowledge on the nature of both quantitative and qualitative job insecurity, little is known regarding the comparative strength of their effects on work-related outcomes. From a handful of studies that simultaneously analyzed the effects of quantitative and qualitative job insecurity on outcomes, three opposing perspectives have emerged. Initially, when considering the severity of these threats (i.e., threat to employment vs. threat to job characteristics), quantitative job insecurity has been perceived as more threatening with stronger consequences to employees’ health and work attitudes [5]. In contrast with that view, recent studies that analyzed quantitative and qualitative job insecurity together suggest that the strength of these effects is either similar [9] or varies depending on the measured outcome variables [6,14,15]. Furthermore, when we look at the reports of the bivariate correlations, the association between qualitative job insecurity and the outcomes seems to be stronger than the one between quantitative job insecurity and the outcomes [16,17]. To address these inconsistencies, the current study simultaneously examined the longitudinal effects of quantitative and qualitative job insecurity, including a wide range of the outcome variables, classified as job strains, and psychological and behavioral coping reactions.

Taking a step further, we took a closer look into the causal order between these two dimensions of job insecurity. Although both quantitative and qualitative job insecurity have been linked with organizational and workplace changes, no previous research examined which dimension of job insecurity is experienced first or how they influence each other over time. A good understanding of the onset and the relationship between the dimensions of job insecurity could facilitate the early recognition of job insecurity among employees and improve the organizational strategies that aim to reduce the consequences of these stressors for employees and organizations. In the current study, we directly addressed these issues by exploring the temporal order between both dimensions of job insecurity in the broader context of the job insecurity–outcomes relationship. We implemented two theoretical streams to explain the associations between quantitative and qualitative job insecurity. First, we considered the Job Insecurity Integrated Model (JIIM) proposed by Chirumbolo et al. (2017) [18] as it suggests that the effects of quantitative job insecurity on the outcomes are fully mediated by qualitative job insecurity. Building on Jahoda’s deprivation theory, the JIIM argues that the threat to job loss directly implies a threat to characteristics of that job, which raises psychological distress and results in job strain and withdrawal reactions [18]. Second, we further extended the JIIM, and on the grounds of COR theory, we proposed an alternative mediational path [19]. Specifically, we argued that the chronic threats to valued job features deplete employees from their resources, which leaves them more vulnerable to threats of job loss. This suggests that the qualitative job insecurity–outcomes relationship is mediated by quantitative job insecurity. We integrated these two frameworks and suggested a reciprocal relationship between quantitative and qualitative job insecurity. In addition, we simultaneously tested both mediation mechanisms (dual-mediation model) to examine the relative importance of each mediator. 

Our study contributes to the literature in three ways. First, we examined the simultaneous effects of quantitative and qualitative job insecurity on a wide range of outcomes, which adds to the understanding of the relative importance of each dimension of job insecurity. By implementing the outcomes classified as job strains and both psychological and behavioral coping reactions, we provide valuable information on whether the importance of a particular dimension of job insecurity is related to the specific outcome under consideration. Furthermore, we controlled for the effects of one dimension of job insecurity while estimating the effect of the other dimension, thus we obtained a more robust estimation of the effects of each dimension of job insecurity. Second, while maintaining the job insecurity–outcomes context, we assessed the relationship between quantitative and qualitative job insecurity. We proposed a theoretical research model to account for the reciprocal relationship between the two dimensions of job insecurity. In doing so, we simultaneously reanalyzed the Chirumbolo et al.’s JIIM and test for an equally plausible opposing mediation process. We further explored the complexity of the relationship between the two dimensions of job insecurity and contrasted the strengths of two possible mediation processes in explaining the job insecurity–outcomes relationship. Third, we addressed the limitations of the previous cross-sectional research by implementing a three-wave longitudinal research design, which allowed us to control for the previous levels of the outcome variables and to examine the temporal order in the mediation processes. 

### 1.1. Job Insecurity, and Its Association with Job Strain and Coping Reactions 

Job insecurity is defined as an individually perceived threat to the continuity of one’s job in the future [5]. Currently, the most widely adopted definition distinguishes two dimensions of job insecurity. First, *quantitative job insecurity* refers to perceived threat to the job as such: employees’ fear they might lose their job. Second, *qualitative job insecurity* defines employees’ perceived threat to the loss or negative change to valuable job features, such as career opportunities, optimal working conditions, or income development [6]. The threat may be appraised cognitively, as a likelihood of loss or negative change, or affectively as a fear or worry. Job insecurity is therefore a *subjective* experience that arises as a result of an individual evaluation of the workplace environment [13]. Some employees may experience high levels of job insecurity within a stable and secure work environment. At the same time, others might feel secure while being confronted with an actual threat to the continuity of their job. As a result, employees from the same work environment may, to a certain degree, experience different levels of job insecurity [20]. A fundamental characteristic of job insecurity is *uncertainty* about the future [21]. That is, employees who feel insecure are not informed about the future of their work, hence they only suspect that changes might occur. Anticipation of negative changes or losses have been shown to be equally or even more detrimental than actual job loss [22,23,24]. 

In line with the stressor–strain perspective, prolonged uncertainty regarding one’s job situation is identified as a prominent work stressor causing detrimental effects to individual and organizational wellbeing (see [7,25,26] for extensive overviews and meta-analyses). The distinction between quantitative and qualitative job insecurity raises the question of the relative salience of the effect of each dimension on the negative outcomes. At first, research on job insecurity mostly focused on the detrimental effects of quantitative job insecurity, since the dimension has been perceived to be more problematic than qualitative job insecurity [5]. A plethora of research, both cross-sectional and longitudinal, has found quantitative job insecurity to be associated with health complaints, negative work attitudes, and a decrease in job performance. At the same time, a growing field of research on qualitative job insecurity has found it to be linked with a deterioration in employees’ wellbeing [11,27,28]. These results conclusively present quantitative and qualitative job insecurity as severe work stressors with detrimental effects to both employees and organizations. In line with this, in the current study we expected to observe a direct negative association between both dimensions of job insecurity and the measured outcomes. 

Despite strong evidence for the severity of the effects of quantitative and qualitative job insecurity, results from comparative studies are inconclusive. On the one hand, De Witte et al. (2010) [9] found no important differences in the strength of the effect of quantitative and qualitative job insecurity on wellbeing and health related outcomes. Both dimensions, with almost equal strength, were positively related to job dissatisfaction, burnout, psychological distress, and psychosomatic complaints. On the other hand, several studies have argued that the strength of the relationship is conditional upon the specific outcome. Given the severity of the threat, quantitative job insecurity might evoke stronger stress reactions and consequently more detrimental health outcomes. At the same time, qualitative job insecurity, which is a threat of loss or negative change to job characteristics, could be linked with negative change in the attitudes toward the job and the organization itself. Indeed, Hellgren et al. (1999) [6] found that quantitative job insecurity had stronger associations with physical and mental health outcomes, while qualitative job insecurity primarily affected work attitudes such as job satisfaction and turnover intention. Furthermore, Tu et al. (2019) [15] found that quantitative job insecurity was more related to employees’ stress-related responses, whereas qualitative job insecurity was more predictive of lower work engagement. In line with current knowledge, we expected that both quantitative and qualitative job insecurity had a negative effect on the outcomes. However, no specific hypothesis regarding the comparative strength and importance of these relationships were formulated in the current study.

Job insecurity has been shown to broadly affect varying aspects of employees’ mental and physical health, work attitudes, and performance. In order to ease the interpretation of our results, we followed the dual classification of these outcomes [29,30]. First, physical and mental health outcomes, which are reactions to stressful situations, were identified as strain. Secondly, work-related attitudes and behaviors that are directed at dealing with a stressful situation were labelled as coping reactions (also, in the job insecurity literature typically known as withdrawal reactions). In the present study, we identified exhaustion, emotional impairment, and cognitive impairment as work-related strain reactions. All three indicate an employee’s inability to perform and constitute core symptoms of burn-out [31]. Secondly, psychological coping reactions can be manifested by low levels of job satisfaction and work engagement and increased levels of turnover intention. These variables reflect employees’ evaluation of the job and reactions aimed at reducing the impact of work stressors, such as job insecurity [30]. Additionally, we included self-rated performance identified as a behavioral coping reaction. Following Campbell’s model of job performance, we classified three types of behaviors: in-role performance (job tasks that are part of the job description), extra-role performance (behaviors that are out of scope of the job description and which help to reach organizational goals), and counterproductive behavior [32]. In sum, we predicted over time associations between both dimensions of job insecurity, strains, and coping reactions as follows:

**Hypothesis** **1.**
*On the relationship between job insecurity and work-related strain: Quantitative and qualitative job insecurity have a positive effect on exhaustion (H1a), emotional impairment (H1b), and cognitive impairment (H1c) over time.*


**Hypothesis** **2.**
*On the relationship between job insecurity and attitudinal coping reactions: Quantitative and qualitative job insecurity have a negative effect on job satisfaction (H2a) and work engagement (H2b) over time, and a positive effect on turnover intention (H2c) over time.*


**Hypothesis** **3.**
*On the relationship between job insecurity and behavioral coping reactions: Quantitative and qualitative job insecurity have a negative effect on in-role performance (H3a) and extra-role performance (H3b) over time, and a positive effect on counterproductive behavior (H3c) over time.*


### 1.2. On the Interrelationship between Quantitative and Qualitative Job Insecurity

Along with the question on the relative importance of each dimension of job insecurity, a debate has arisen over how those two components are interrelated. To date, research on that issue is almost non-existent. Studies that included quantitative and qualitative job insecurity in the same analysis provide indirect evidence that both dimensions are positively related [9,15,18,33,34]. However, no previous research has examined the associations over time between these two dimensions. Disentangling the order, the direction, and the strength of those relations might provide insights into the process of the development of job insecurity and help to shed light on their compound effects on the outcomes. In line with theory and previous empirical findings, we proposed that quantitative and qualitative job insecurity form a complex reciprocal relationship, including direct causation (from quantitative job insecurity to qualitative job insecurity) and reverse causation (from qualitative job insecurity to quantitative job insecurity).

#### 1.2.1. Quantitative Job Insecurity to Predict Qualitative Job Insecurity

According to Jahoda’s latent deprivation theory, employment provides access to unique resources [35]. Apart from financial stability, being employed grants diverse latent benefits such as: opportunity for interaction with peers (*social contact*), daily schedule and purpose (*time structure*), social recognition and status (*status/identity*), engagement in specific job-related tasks (*enforced activity*), and, lastly, opportunity for a meaningful contribution to society (*collective purpose*) [35]. As such, losing a job means losing all the benefits that come along with the job. In the context of job insecurity, employees who perceive a threat to their employment (quantitative job insecurity) will also experience a threat to all the benefits that come along with the job (qualitative job insecurity). Stress reactions caused by the threat of losing highly valued work features might in turn explain the negative effects on employees’ health and work attitudes [36,37].

In 2017, Chirumbolo and colleagues proposed the Job Insecurity Integrated Model [18] in which they directly addressed the relationship between quantitative and qualitative job insecurity. In line with Jahoda’s deprivation model, they theorized that quantitative job insecurity cognitively precedes qualitative job insecurity. Furthermore, they argued that in relation with the outcomes, qualitative job insecurity mediates the effects of quantitative job insecurity. Indeed, their results suggested that qualitative job insecurity fully mediated the effects of quantitative job insecurity on mental health and work attitudes. Similar findings were reported by Callea et al. (2019) [38], who extended the outcomes with emotional exhaustion and psychological symptoms.

In line with this theoretical framework and previous research, we proposed that qualitative job insecurity mediates the association between quantitative job insecurity and the outcomes. More specifically, we hypothesized:

**Hypothesis** **4.**
*Based on Jahoda’s deprivation theory, quantitative job insecurity (QN_t-1_) has a positive direct effect on qualitative job insecurity (QL_t_) over time.*


**Hypothesis** **5.**
*Qualitative job insecurity mediates (T_2_) the indirect effects of quantitative job insecurity (T_1_) on the outcomes (T_3_): work-related strains (H5a), psychological coping reactions (H5b), and behavioral coping reactions (H5c).*


#### 1.2.2. Qualitative Job Insecurity to Predict Quantitative Job Insecurity

At the same time, the inverse relation between both dimensions of job insecurity is equally plausible. In accordance with the organizational stress literature, stress among employees usually develops as a complicated sequence of unfavorable events rather than a one-time incident [39]. That said, the threat of losing a job could potentially grow as a consequence of chronic threats to job characteristics spread over time.

According to the conservation to resources (COR) theory “individuals strive to obtain, retain, foster, and protect those things they centrally value” [40] (pp. 103–104) and stress occurs when these resources are either lost or threatened with loss. In the work context, resources include objects (e.g., tools for work), personal characteristics (e.g., self-efficacy), energy resources (e.g., money, knowledge), and conditions (e.g., tenure, type of contract, position on the company). A stable employment with all its characteristics is a set of valuable resources, and a threat to their continuity leads to strains. At the same time, individuals whose resources are threatened are more vulnerable to resource loss and less capable of resource gain [41]. Hence, employees who perceive a threat to valued job characteristics might interpret signals regarding organizational changes as more threatening, leading to negative reappraisals [41]. Consequently, they may perceive neutral or even positive change regarding their job and all its aspects as negative, causing a further increase of job-related insecurity. In line with COR, we formulated the following hypotheses:

**Hypothesis** **6.**
*Based on the conservation of resources theory, qualitative job insecurity (QL_t-1_) has a positive, direct effect on quantitative job insecurity (QN_t_) over time.*


**Hypothesis** **7.**
*Quantitative job insecurity mediates (T2) the indirect effects of qualitative job insecurity (T1) on the outcomes (T3): work-related strains (H7a), psychological coping reactions (H7b), and behavioral coping reactions (H7c).*


### 1.3. Present Study

In the present study, we proposed a theoretically well-grounded research model to address the relationship between quantitative and qualitative job insecurity and their compound effects on the outcomes. Although the two mediation mechanisms proposed in the previous section relate to different theoretical streams, we continued with conservation of resources theory to join both mechanisms in one model (see Figure 1). A unique feature of conservation of resources theory is that it underlines the possibility of reciprocal relationships. First, according to Hobfoll et al. (2018) [40], resources do not develop individually but rather in packs or caravans. More specifically, work-related resources, (e.g., career opportunities) are usually linked with complementary resources (such as access to life-long learning platforms). Through environmental conditions, defined as caravans passageways, the growth and development of these resources is either fostered and protected or undermined and obstructed [42]. Employment and job features are closely related resources. It was expected, then, that the threat to one of them, whether it is an employment (quantitative job insecurity) or a specific threat to job characteristics, such as task significance or career opportunities (qualitative job insecurity), may, over time, affect the other type of job insecurity. Second, resource loss has a spiraling nature, meaning that resource loss engenders future loss. In fact, employees with less resources are more vulnerable to further resource loss. Threat to or lack of resources may thus lead to threat of loss to other, closely related resources, which further boost the negative effects on employees’ wellbeing in the long run. In accordance with this, a change in resources, especially in terms of threat, may provoke a threat to other work-related resources, resulting in the complex reciprocal relationship advocated in this study.

Herewith, we propose a dual mediation model that accounts for the bidirectional relationship between quantitative and qualitative job insecurity. More specifically, we suggest that both dimensions of job insecurity, in addition to the direct effect on the outcomes over time, affect these outcomes indirectly through the other dimension. Thus, we simultaneously re-examined the previously suggested mediation role of qualitative job insecurity and contrasted it with an alternative process where quantitative job insecurity mediates the indirect effects of qualitative job insecurity on the outcomes.

## 2. Materials and Methods

### 2.1. Sample and Procedure

Data for the present longitudinal study were collected from Flemish employees (i.e., the Dutch-speaking region of Belgium). The data were collected as part of a larger study (the authors would like to thank Steffie Desart and Anahí Van Hootegem from KU Leuven for sharing their dataset and providing us with all the information about the data collection process). Researchers published an ad on the website of an online HR magazine (vacature.com) calling on people to participate in a survey on occupational health and wellbeing. Data were collected by means of an online survey using a non-probability sampling method. Respondents were asked to access the questionnaire via a link to an online tool provided in the ad. In the introduction to the survey, researchers clearly stated the purpose of the study and assured voluntary participation and anonymous processing of the data. A total of 2355 participants filled out the questionnaire during the first wave, collected in September 2017 (T1). All interviewees were invited to participate in the subsequent two waves, which took place in March 2018 (T2) and September 2018 (T3), hence a 6-month time lag between each wave. Overall, 1494 employees filled in the questionnaire at T2 (63.4% response rate) and 1114 at T3 (47.3% response rate). To obtain a homogeneous sample for this study and to control for contextual bias, we excluded people who, throughout the observation period, had experienced job transition or who had stopped working altogether (*n* = 352). The final sample included 2003 employees, out of which: 859 (43%) participated in all three waves (T1T2T3); 580 (29%) responded only in the first wave (T1); 326 (16%) completed the survey during the first two consecutive waves (T1T2); 238 (12%) employees filled in the survey during the first and the last wave (T1T3). Multinomial logistic regression was performed to test for an attrition bias (see Appendix A). The results indicated that respondents who presented higher turnover intention had 30% higher odds of dropping out after the first wave (T1 respondents) and 26% higher odds of not responding in the second wave (T1T3 respondents). Furthermore, we observed that respondents with lower qualitative job insecurity had almost 13% higher odds to leave the survey after first wave (T1 respondents). These results indicate that our sample due to drop out may underrepresent employees’ turnover intention and overrepresent employees with higher levels of qualitative job insecurity.

The final sample consisted of 58.4% women (*n* = 1170). The age of participants varied between 20 and 60 years old (*M* = 40.93; *SD* = 10.55). Less than 5% of the participants had a lower secondary education degree, 62.5% had obtained higher secondary or non-university education, and 33% of participants had a university degree (high education: including bachelor, master, and doctorate degrees). Respectively, 6% were blue-collar workers (2% unskilled workers; 4% skilled workers), 61% were white-collar workers (26% lower-level and administrative clerk; 35% middle-level employee), and 33% were in managerial positions (24% low- and middle-level management; 9% senior management). Below 1% of the respondents worked in the primary sector (extraction of raw materials/farming/fishing), 28% worked in the secondary sector (“industry”: manufacturing/utilities), 35% worked in the tertiary sector (“services”: retail/financial services/communication/hospitality/real estate/information technology), and 28% worked in the public sector (“government”: education/public administration/research and development). The majority of the respondents worked in the private sector (80.3%) with a permanent contract (97.2%). Approximately 80% of the interviewees worked full-time. On average, respondents had 10.66 years of positional tenure (*SD* = 9.43). Regarding these sociodemographic variables, the sample was a good representation of the Flemish population (see Appendix B).

### 2.2. Measurements

All variables were measured at three consecutive times using a selection of internationally validated scales. Reliability of the measurement scales was examined with Cronbach alpha for the multi-item scales, and test-retest reliability for the single-item scales.

#### 2.2.1. Job Insecurity

The Job Insecurity Scale (JIS) (developed by [43], validated by [44]) was used to measure *quantitative job insecurity.* This four-item scale measured cognitive (e.g., *“Chances are, I will soon lose my job”*) and affective (e.g., *“I feel insecure about the future of my job”*) aspects of the construct. The items were rated on a five-point Likert scale from 1 (*totally disagree*) to 5 (*totally agree*). The internal consistency was α = 0.92 for T1, α = 0.94 for T2, and α = 0.93 for T3.

*Qualitative Job Insecurity* was measured with a four-item scale (developed by De Witte and De Cuyper; used in the previous studies [16,45]). The scale captured cognitive (e.g., *“I think my job will deteriorate in the near future”*) and affective (e.g., *“I am worried about how my job will look in the future”*) aspects of employees’ insecurities over job characteristics without listing specific job features. The items were rated on a five-point Likert scale from 1 (*totally disagree*) to 5 (*totally agree*). The internal consistency was α = 0.90 for T1, α = 0.92 for T2, and α = 0.91 for T3.

#### 2.2.2. Job Strains

Job strains were identified as the core symptoms of burnout and were measured using a burnout assessment tool (BAT) [31]. The first dimension, *exhaustion* was measured with three items that refer to a severe loss of energy resulting in physical and mental exhaustion (e.g., *“At work, I feel mentally exhausted”*). The second dimension, *emotional impairment* was a three-item measure of intense emotional reactions and overwhelming feelings at work (e.g., *“At work, I feel unable to control my emotions”*). Finally, *cognitive impairment* was a three-item measure of subjectively assessed memory problems, attention/concentration deficits, and poor cognitive performance (e.g., *“At work, I have trouble staying focused”*). Respondents rated these items on a five-point Likert scale from 1 (*never*) to 5 (*always*). The internal consistency was α = 0.90 at T1, α = 0.90 at T2, and α = 0.89 at T3 for exhaustion; α = 0.88 at T1, α = 0.90 at T2, and α = 0.87 at T3 for emotional impairment; α = 0.91 at T1, α = 0.90 at T2, and α = 0.91 at T3 for cognitive impairment.

#### 2.2.3. Psychological Coping Reactions

The three-item UWES-3 scale was used to measure the three dimensions of work engagement: vigor (*“At my work, I feel bursting with energy”*), dedication *(“I am enthusiastic about my job”*), and absorption (*“I am immersed in my work”*) [46]. The items were rated on a five-point Likert scale from 1 (*never*) to 5 (*always*). The Cronbach alphas for the scale were α = 0.86 at T1, α = 0.84 at T2, and α = 0.85 at T3.

*Turnover intention* was measured with a one-item scale derived from the online questionnaire *Energy Compass* [47]. The item was designed to measure the extent to which an employee is planning to change jobs in the following year. Respondents were asked to rate this statement on a 5-point Likert scale from 1 (*totally disagree*) to 5 (*totally agree*). Test–retest reliability of the measurement was examined with the intraclass correlation coefficient (ICC), which values between 0.5 and 0.75, and 0.75 and 0.9 indicate moderate and good reliability, respectively [48]. The average measure ICC was 0.817 with a 95% confidence interval from 0.817 to 0.854 (F(1030,2060) = 6.228, *p* < *0*.001). Hence, the measurement of turnover intention presented good test–retest reliability.

*Job satisfaction* was assessed by means of a single-item measure [49]. Respondents were asked to rate their overall job satisfaction on a scale from 1 (*very unsatisfied*) to 10 (*very satisfied*). The average measure ICC was 0.743 with a 95% confidence interval from 0.715 to 0.768 (F(1101,2202) = 3.918, *p* < *0*.001). Test–retest reliability of the measurement showed a moderate reliability of job satisfaction.

#### 2.2.4. Behavioral Coping Reactions

Employees’ performance was assessed using the online questionnaire *Energy Compass* [47]. Two constructs were measured: in-role and extra-role performance. First, *in-role performance* was measured with three items that assessed the extent to which employees fulfilled the duties required by the job (e.g., *“I meet all the requirements that my position places on me”*). The three-item measure of *extra-role performance* examined the frequency of positive behaviors, which do not fit a formal job description (e.g., *“I help my colleagues with their work when they return from a period of absence”*). Respondents were asked to rate these items on a 5-point Likert scale from 1 (*never*) to 5 (*always*). The internal consistency was α = 0.86 at T1, α = 0.88 at T2, and α = 0.85 at T3 for in-role performance and α = 0.78 at T1, α = 0.77 at T2, and α = 0.75 at T3 for extra-role performance.

*Counterproductive behavior,* defined as an employee’s intentional behavior that harms or intends to harm the organization, was measured with a four-item scale [50]. Participants were asked to evaluate, on a scale from 1 (*never*) to 5 (*always*), how often in the last six months they had shown specific behavior, like taking longer breaks or not following the boss’s instructions (e.g., *“Taking material from the work home without permission for personal use”*). The Cronbach alphas for the scale were α = 0.66 at T1, α = 0.63 at T2, and α = 0.64 at T3.

#### 2.2.5. Control Variables

We considered three potentially relevant control variables, including gender (0 = male; 1 = female), positional tenure (years), and level of education (1 = primary education; 2 = lower secondary education; 3 = higher secondary education; 4 = non-university higher education; 5 = university higher education; 6 = doctorate), treated as a continuous variable that represents a range from less educated to highly educated. First, according to job dependence theory, male employees experience higher economic insecurity [5], which translates into higher perceived job insecurity, as they feel more responsible to provide financial stability for their family [22,51]. Second, human capital theory explains that more educated employees with longer tenure exhibit more positive work attitudes and behaviors. Empirical evidence shows that higher education and longer tenure grants access to better jobs with higher salaries and additional resources, which results in higher job satisfaction and task performance, and gives more incentives to remain in an organization [52,53,54].

Despite this theoretical rationale and the evidence from previous research, the examination of the bivariate correlations (see Table 1) showed no significant associations between gender and the two dimensions of job insecurity. Furthermore, we found no significant correlation between education and job satisfaction or performance. On the other hand, consistent with our theory-based expectations, positional tenure was negatively correlated with turnover intentions. Hence, to facilitate the interpretation of the results and to maximize statistical power we performed the analysis without controlling for gender and level of education. However, given the significant bivariate corrections and our theoretical rationale, we controlled for positional tenure.

### 2.3. Analysis

To address the research questions, we conducted structural equation modelling using the Lavaan package in R software [55]. We followed the step-wise procedure outlined by Cole and Maxwell (2003) [56] and Little et al. (2007) [57] (see [27,28,58] for a similar methodology). Preliminary data analysis on multicollinearity (i.e., bivariate correlations higher that r = 0.85) and nonnormality (i.e., extreme values, above 3.0 for skewness and 10.0 for kurtosis), indicated no violations [59]. To address substantial attrition throughout the study, we implemented the full information maximum likelihood (FIML) estimator, which has been shown to be a superior method in dealing with missing data to produce unbiased parameter estimates [60,61].

Model fit was evaluated using several goodness-of-fit indices, specifically Chi-square (χ^2^), comparative fit index (CFI), the Tucker–Lewis index (TLI), the root mean square error of approximation (RMSEA), and the standardized root mean square residual (SRMR) [59,62]. Considering a sensitivity of χ^2^ to the sample size, we followed Hu and Bentler’s (1999) [63] recommendations and considered the following criteria for a good model fit: values higher than 0.95 for CFI and TLI, and lower than 0.06 and 0.08 for RMSEA and SRMR, respectively. Furthermore, alternative models were compared based on ΔCFI and ΔRMSEA, where a change of ≤−0.01 and ≤0.015, respectively, indicates a better model fit [64,65].

First, we conducted a confirmatory factor analysis (CFA) to evaluate the measurement model fit of the hypothesized 33-factor model (M1), in which 96 items that measured quantitative job insecurity, qualitative job insecurity, and nine outcome variables were loaded on their respective latent factors at every time point (32 items at each measurement wave). We allowed the measurement errors for each item to covary across time. Next, we compared that model with three alternative models: a 15-factor model (M2), in which we merged the outcome items to load on three large factors: job strains, psychological coping reactions, and behavioral coping reactions; a 12-factor model (M3), in which the items that measured quantitative and qualitative job insecurity loaded on one general job insecurity factor; and a 3-factor model (M4), in which all items loaded on one factor at every time point.

In the following step, we evaluated longitudinal measurement invariance to test whether the respective items represented the same underlying constructs over time [57]. The best fitting measurement model chosen from a previous sequence was used as the initial configural invariance model (equal factor structure across time). Subsequently, we fitted a sequence of more restricted (and nested) models to test the validity of the imposed constraints. The baseline model was compared with a metric invariance model (M5), which had equality constraints placed on factor loadings of the corresponding indicators across time. The latter was then compared to a strong invariance model (M6), in which in addition to the loadings, the intercepts of the corresponding items, were constrained to be equal across time. In the final step, we evaluated if a strict measurement invariance held (M7), in which the residual variances of the corresponding items were equated across time. Research indicates that metric invariance is a minimum requirement to proceed with the evaluation of the structural paths of direct and mediated effects among latent factors [66].

Subsequently, we estimated and compared the model fit of four structural models in order to select the best model to test the hypothesized mediation effects. In this step, we added positional tenure as a control variable to each of the estimated models. Firstly, we estimated a structural model with autoregressive paths (M8). Building on that model, we then estimated the longitudinally extended Chirumbolo et al.’s JIIM with qualitative job insecurity as a mediator (M9). Afterward, we analyzed the reversed mediation model with the mediating role of quantitative job insecurity (M10). Lastly, we fitted the hypothesized dual-mediation model (M11) that integrates the reciprocal relationships between quantitative and qualitative job insecurity. This procedure allowed us to test whether the model proposed in the current study fit the data better than the alternative models with an estimated unidirectional relationship between quantitative and qualitative job insecurity. In the final step, following the recommendations of Cole and Maxwell (2003) [56] we tested whether the best-fitting structural model was invariant across time. First, we fixed the autoregressive paths to be equal across time (M12) and compared them with the baseline model. Subsequently, we added equality constraints on cross-lagged paths from a predictor to a mediator (*paths a*) (M13), followed by a model with constrained cross-lagged paths from a mediator to the outcome variables (*paths b*) (M14). The model with the best fit was then used to estimate the statistical significance of mediation effects. We used a bootstrapping method (5000 resamples) to calculate the 95% confidence intervals for the indirect effects.

## 3. Results

### 3.1. Descriptive Statistics

Table 1 shows the means, standard deviations, correlations, and reliabilities of all variables. The mean values for the job insecurity dimensions and outcomes were relatively stable across time. Low standard deviations indicate small variations between the participants. We also observed that participants on average experienced higher qualitative than quantitative job insecurity. Quantitative and qualitative job insecurity were significantly related to the outcome variables, as expected. Of note, qualitative job insecurity correlated stronger with the outcome variables across all observation points. Among the dependent variables, positive correlations were found between job strains, turnover intentions, and counterproductive behavior as well as work attitudes with job performance; negative correlations were found for job strains with work attitudes and job performance as well as for work attitudes with turnover intentions and counterproductive behavior.

### 3.2. Measurement Model and Measurement Invariance

Table 2 presents fit indices for the models with a competing factorial structure of the measurement model. The hypothesized 33-factor model (M1) showed a good fit to the data (*χ*^2^(3852) = 7246.195, CFI = 0.967, TLI = 0.961, RMSEA = 0.021, SRMR = 0.043). As indicated by the Δ*χ*^2^ difference test, the alternative 15-, 12-, and 3-factor models presented a significantly worse fit to the data (Δ*χ*^2^(417) = 17,027.69, *p* < 0.001; Δ*χ*^2^(456) = 24,317.11, *p* < 0.001; Δ*χ*^2^(519) = 41,809.21, *p* < 0.001, respectively). Therefore, the hypothesized 33-factor model was preferred for further analysis.

Subsequently, we investigated longitudinal measurement model invariance. A Chi-square difference test indicated that all constrained models showed significantly worse fit than the configural model, which suggested non-invariance. However, the large sample size (N = 2003) might have biased Δ*χ*^2^ results against invariance. Therefore, following Chen (2007) [67], we applied ΔCFI < 0.01 and ΔRMSEA < 0.015 thresholds as a criterion for measurement invariance. Subsequent models, with gradually added constrains, met the measurement invariance criterion, and did not decreased the model fit (see Table 2). The strict invariance model (M7; the model with equality constrains on the factor loadings, intercepts, and error variances) showed a good model fit (*χ*^2^(4014) = 7838.118, CFI = 0.963, TLI = 0.958, RMSEA = 0.022, SRMR = 0.043) and met the criterion for measurement invariance (ΔCFI = 0.003; ΔRMSEA = 0.001). Hence, we concluded that the measurement model was invariant across time and proceeded with the analysis of the structural model.

### 3.3. Structural Model and Stability of the Model

In order to select the best model to test the hypothesized cross-lagged relationships, we compared four structural models. We added positional tenure as a covariate to all competing models and controlled for its effect on the modelled variables at the second measurement time [57]. Table 3 presents the overview of the results. The mediation model proposed by Chirumbolo et al. (2017) [18], with qualitative job insecurity in a role of mediator (M9), fit the data significantly better than the autoregressive model (Δ*χ*^2^(29) = 132.45, *p* < 0.001). Similarly, the reversed mediation model (M10) showed a significant model fit improvement (Δ*χ*^2^(29) = 60.72, *p* < 0.001). The analysis of the path coefficients showed that both models included significant and complementary pathways. The final model combined Chirumbolo et al.’s JIIM and the alternative reverse mediation model. The hypothesized dual-mediation model (M11) showed acceptable model fit (*χ*^2^(4522) = 11,606.381, CFI = 0.931, TLI = 0.928, RMSEA = 0.028, SRMR = 0.072) and significantly better fit than Chirumbolo et al.’s JIIM (Δ*χ^2^*(29) = 58.928, *p* < 0.001). Therefore, the dual-mediation model, with the hypothesized reciprocal relationships between quantitative and qualitative job insecurity, was chosen for the subsequent series of analyses.

Accordingly, we examined the stability of the model. Comparably to the estimation of measurement model invariance, we employed the 0.01 and 0.015 thresholds for ΔCFA and ΔRMSEA, respectively, as a criterion for structural model invariance across time. First, we put equality constrains on the autoregressive paths (M12), which did not significantly decrease model fit (ΔCFA = 0.001; ΔRMSEA = 0). Additional equality constraints on the paths from the predictors to the mediators (M13) did not worsen the model fit (ΔCFA = 0; ΔRMSEA = 0). The final model, with additional equality constrains on the paths from the mediators to the outcome variables (M14), presented a good fit to the data (*χ*^2^(4553) = 11,663.687, CFI = 0.931, TLI = 0.928, RMSEA = 0.028, SRMR = 0.072) and was not significantly worse than the partially constrained model (ΔCFA = 0; ΔRMSEA = 0). Thus, the relationships over time between the constructs were invariant across time, and we proceeded with that model to examine the hypothesized effects.

### 3.4. Test of the Hypotheses

The final model and all estimated path coefficients were summarized in the figure included in Appendix C. Figure 2 presents the significant path coefficients. The results showed that qualitative job insecurity has a direct positive effect on the symptoms of burnout (exhaustion, emotional impairment, and cognitive impairment) six month later, while controlling for the effects of quantitative job insecurity and previous levels of the outcome variables. The direct effect of quantitative job insecurity on the burnout symptoms was found not statistically significant. Thus, Hypotheses 1a, 1b, and 1c, which assumed positive direct effects of job insecurity on job strains over time, were only partially supported. As expected, we found qualitative job insecurity to have a direct negative effect on job satisfaction and work engagement and a positive effect on turnover intention, while the effects of quantitative job insecurity were not statistically significant. These results partially support Hypotheses 2a, 2b, and 2c, which proposed negative associations between quantitative and qualitative job insecurity over time with psychological coping reactions. In contrast to Hypotheses 3a and 3b, the results did not support a significant relationship between job insecurity and job performance over time. Quantitative and qualitative job insecurity had no direct effect on in-role performance and extra-role performance six months later. Finally, qualitative job insecurity was associated with an increase in counterproductive behavior six months later, therefore we found partial support for Hypothesis 3c.

The analysis of the relationship between quantitative and qualitative job insecurity over time showed that, in contrast with the Chirumbolo et al.’s JIIM [18], we found no direct effect of quantitative job insecurity on qualitative job insecurity six months later. Thus, Hypothesis 4 was rejected. These findings resulted in the rejection of Hypothesis 5, which assumed that the effects of quantitative job insecurity (T_1_) over time on the outcomes (T_3_) are mediated through qualitative job insecurity (T_2_). On the other hand, we found qualitative job insecurity to have a positive direct effect on quantitative job insecurity six months later, while controlling for the previous levels of quantitative job insecurity. Thus, the results support Hypothesis 6. However, since quantitative job insecurity was found to have no effect over time on the outcomes, Hypothesis 7, in which quantitative job insecurity acts as a mediator between qualitative job insecurity and the outcomes, was rejected.

## 4. Discussion

In the current study, we analyzed the longitudinal associations between quantitative job insecurity, qualitative job insecurity, and their negative outcomes. First, we aimed to replicate and extend previous research by investigating the joint effect of quantitative and qualitative job insecurity over time on a range of outcomes. Contrary to the expectations, we found that once analyzed together, only qualitative job insecurity had a detrimental effect on employees’ wellbeing by intensifying job strains (exhaustion, emotional impairment, cognitive impairment), negative work attitudes, and behaviors (job dissatisfaction, work disengagement, turnover intention and counterproductive behavior), whereas quantitative job insecurity did not affect any outcome variable after a six-month period. Thus, we only partially confirmed our hypotheses on the job insecurity-outcomes relationship.

Secondly, we aimed to investigate the nature of the relationship between both dimensions of job insecurity over time. We argued that quantitative and qualitative job insecurity form a complex bidirectional relationship. In addition to the direct effects on the outcomes, we also expected to find indirect effects of each dimension of job insecurity. That is, we expected that the relationship between one dimension of job insecurity (either quantitative or qualitative job insecurity) and the outcomes is partially mediated by the other dimension. However, the results only revealed a stable unidirectional relationship of qualitative job insecurity on quantitative job insecurity six month later. We did not find mediation processes when predicting outcomes over time.

### 4.1. Theoretical Implications

The first contribution of this study has been the simultaneous examination of both quantitative and qualitative job insecurity as work stressors, which allowed us to compare the strength of their effects on various outcomes. To date, comparative research on the concurrent effects of quantitative and qualitative job insecurity are inconclusive. While at first scholars concluded quantitative job insecurity to be a more severe work stressor [5], [22], further research found that either both dimensions pose similar threats [9] or the strength of the effects of each dimension of job insecurity depends on the measured outcomes [6,14,15]. Amid those contrasting findings, we expected to find negative effects of both dimensions without specifying differences in the strength of the associations. Interestingly, the results only partially supported our hypotheses and, more importantly, did not align with any of the earlier mentioned conclusions. Specifically, we found that qualitative job insecurity was associated with the majority of the measured outcomes, that is: core burnout syndromes (exhaustion, emotional and cognitive impairment), work attitudes (work engagement, job satisfaction, turnover intention), and counterproductive behavior. At the same time, quantitative job insecurity showed no associations with any of the measured outcomes over time. In other words, when analyzed together, only the threat to job characteristics predicted a negative change to employees’ mental health, work attitudes, and job performance six months later. Thus, our results add to the ongoing debate and suggest a fourth possibility; that is, the impact of qualitative job insecurity on employees and organizations is more severe when compared to the impact of quantitative job insecurity.

It is possible that, when analyzed together, qualitative job insecurity explains the variance in the measured outcomes, which in a separate analysis would be concluded as a result of a direct effect of quantitative job insecurity. We found two reasons for that explanation. First, qualitative job insecurity defines a perceived threat to the future work conditions without specifying the exact work features. In comparison with quantitative job insecurity, which is a specific measurement of the perceived threat to job loss, this dimension captures a broad scope of job-related insecurities that employees might experience, which ultimately makes the qualitative dimension of job insecurity explain more variance in change in the outcome variables. Second, as previous research suggests, the threat of job loss is an ultimate threat to the work conditions, but not the other way around [27,36]. Hence, when analyzed together, the threat of job loss could be partially captured as a threat of change to the future work conditions, which could explain why qualitative job insecurity was found to have a stronger impact on the measured outcomes. Considering that the current study was the first to simultaneously estimate the longitudinal effects of quantitative and qualitative job insecurity on the outcomes while controlling for the effects of the other dimension of job insecurity, the conclusions need to be taken with caution. Further research is needed to support these findings and to provide more insight into the nature of these joint effects.

Furthermore, we acknowledge that the response to quantitative and qualitative job insecurity could be affected by the cultural values of our sample. Considering two dimensions of a culture, namely uncertainty avoidance and performance orientation, it is plausible that Flemish employees can deal particularly well with quantitative job insecurity while being more vulnerable to the impact of qualitative job insecurity.

Uncertainty avoidance defines the extent to which members of a particular culture feel threatened by unknown future situations [68]. In order to avoid or reduce negative outcomes of these unpredictable or unforeseen situations, cultures high on uncertainty avoidance develop sets of social norms and well-organized procedures to deal with uncertainty. More specifically, Flanders, which scores high on the uncertainty avoidance index (UAI), has developed well-established institutions and policies to decrease the unpredictability of job loss (e.g., dominant permanent contracts; see Appendix B) and to tackle the negative outcomes of job loss (e.g., social safety net) [69]. Thus, employees who lose their job can rely on governmental help to keep financial liquidity and to look for a new job. Although research on the effect of cultural dimensions on the consequences of experienced quantitative and qualitative job insecurity is scarce, an interesting study by Sender et al. (2017) [70] examined the moderating role of cultural dimensions in different regions of Switzerland on the associations between the two dimensions of job insecurity and negative outcomes. Interestingly, they found that the link between quantitative job insecurity and turnover intention was stronger among employees from the French-speaking region of Switzerland, which is lower on uncertainty avoidance than in the German-speaking region of Switzerland. It is possible that, although Flemish employees do experience threats of job loss, the simultaneous awareness that the government protects their citizens from the consequences of job loss reduces the negative effects of this work stressor on employees’ health and work attitudes.

Whereas a high score on uncertainty avoidance could lead Flemish society to organize in a way that reduces the negative effects of quantitative job insecurity, a performance orientation might explain particularly long-lasting reactions to the threat to job characteristics. The cultural dimension performance orientation defines cultures that value life-long training and education as essential for a successful work life. Individuals from such cultures believe they are in control of their career paths, more often display initiative, and are rewarded for their achievements [71]. Thus, workplace changes that threaten those pursuits might be particularly damaging for employees within performance-oriented cultures. Indeed, in the same analysis, Sender et al. (2017) [70] found that in the German-speaking region of Switzerland, which scored higher on the performance-oriented dimension than the French-speaking region, the link between qualitative job insecurity and job satisfaction was stronger. To our knowledge, Flemish society has yet to be measured on the performance orientation dimension. However, based on the other cultural dimensions, such as a high score on individualism and an intermediate score on masculinity [72], we can assume that Flanders scores moderate-to-high regarding a performance-oriented culture, which might explain the significant importance of qualitative job insecurity in predicting the change in outcome variables. In line with this reasoning, we emphasize the importance of the link between cultural dimensions and the response to quantitative and qualitative job insecurity. Future research could contribute to the literature with a cross-cultural study to examine the effect of Hofstede’s cultural dimensions on the relationship between the dimensions of job insecurity and various outcomes. For example, subsequent study could examine whether employees from Wallonia, the French-speaking region of Belgium, do react differently to both quantitative and qualitative job insecurity than employees from Flanders, and whether this difference is related to cultural differences between those regions. 

The second contribution of this study has been the examination of the associations and the temporal order of quantitative and qualitative job insecurity in the context of the job insecurity–outcomes relationship. As such, the current study was the first to answer the call for a longitudinal examination of Chirumbolo et al.’s JIIM, which proposed qualitative job insecurity as a mediator of the relationship between quantitative job insecurity and outcomes [18]. We further expanded on that model and additionally proposed an equally plausible alternative mediating process. In line with COR theory, we argued that quantitative job insecurity is preceded and directly affected by the alleviated threats to job characteristics. Subsequently, we expected to find a reciprocal relationship between quantitative and qualitative job insecurity. Our results only partially supported the proposed model. In contrast with previous research, we found no significant effect of quantitative job insecurity on qualitative job insecurity over time. Hence, we did not confirm the mediating role of qualitative job insecurity, as proposed in the JIIM. On the other hand, qualitative job insecurity was associated with an increase in quantitative job insecurity six months later. Overall, our results suggest an opposing view to the one proposed in Chirumbolo et al.’s JIIM: when examined longitudinally, qualitative job insecurity precedes and leads to quantitative job insecurity. These results were in line with conservation of resources theory. First, a threat to resources is a cyclic process in which initial threats engender future threats. Furthermore, closely related resources travel in caravans, meaning that threats to a particular job resource might elicit the threat of others. In line with these corollaries, we concluded that employees who experience alleviated threats to their job characteristics interpret all information regarding anticipated workplace changes as threatening, which further intensifies their perception of insecurity. Over time, these threats spread on to other work resources. Employees might then begin to question the security of their overall employment, which invokes perceived quantitative job insecurity [39].

Additionally, various reasons might account for some unexpected findings. First, the failure to longitudinally confirm JIIM could potentially be due to the difference in the operationalization of qualitative job insecurity [26]. The growing amount of research on qualitative job insecurity resulted in a plurality of instruments that cover different aspects of the construct [6,73,74]. Whereas Chirumbolo et al.’s JIIM used Hellgren et al.’s (1999) [6] scale to measure threat of loss regarding four pre-specified job features (career and wage development as well as future prospects and task stimulation), we used a short and context-independent scale specifically developed to examine the extent of employees’ perceived insecurity without the reference to specific job characteristics. By implementing such a generic scale, we covered more aspects of qualitative job insecurity, which might have altered the relationship between the dimensions. In order to control for these differences, future research could longitudinally re-examine JIIM with the exact measurement scales for quantitative and qualitative job insecurity that were used in the original study.

In addition, it is possible that the relationship between quantitative and qualitative job insecurity is still reciprocal but that the effects of each dimension on the other occur at different time lags. In other words, whereas we observed that the threat of job characteristics was associated with an increase in threats to job loss six months later, the effect of threats to job loss on threats to job characteristics over time might only be observed once a different time lag is applied. When analyzed longitudinally, the length of the time lag between the measurement waves needs to be properly estimated in order to observe the underlying temporal order [75]. In the current study, the time lag of six months might have been too long to observe the effects of quantitative job insecurity on qualitative job insecurity. In line with Jahoda’s deprivation model, we can expect that an increase/decrease in the threat to job loss almost synchronously results in an increase/decrease in the threat to valued job features. Hence, it is possible that this immediate reaction can be observed only cross-sectionally or with relatively short time lags of a few hours or days [76].

### 4.2. Practical Implications

This study particularly emphasized the importance of qualitative job insecurity. Threats to job characteristics not only lead to negative consequences for employees’ health, work attitudes, and performance, but also engender the threat of job loss six months later. A focal point of HRM practices should be the implementation of new practices that primarily aim at reducing the experience of qualitative job insecurity among employees and buffer its effect on the outcomes. This could be achieved by increasing investments in career development, which not only elicit engagement in the organization but also send a clear signal on a secured future role in the organization. Furthermore, organizations should establish clear formal communication channels to address prospective workplace changes, which have been shown to reduce employees’ feelings of job insecurity [77]. Finally, negative outcomes of qualitative job insecurity might be reduced if organizations create opportunities for employees to participate in decision-making processes regarding workplace changes that directly affect the future characteristics of their job [78]. The antecedental role of qualitative job insecurity in the development of quantitative job insecurity suggests that these practices might also indirectly decrease the experience of quantitative job insecurity and may thus hamper its negative effects.

### 4.3. Limitations and Future Research

The current study had several limitations that should be acknowledged. First, data were collected via a non-probability sampling procedure, which might have resulted in sampling bias. More specifically, access to the survey was restricted solely to the readers of the HR online magazine vacature.com (accessed on 1 July 2017), hence certain categories of the Flemish working population might be overrepresented. To test for this, we compared the sample demographics with those of the Flemish employed population (see Appendix B). Our sample was roughly representative of the employed Flemish population based on gender, age, education, type of contract (permanent vs. temporary), work timeframe (full-time vs. part-time), and sector (public vs. private). However, the distribution of other characteristics of the employed population in Flanders, for example job position, specific work sector, or the size of the company, could not be compared. These characteristics were commonly identified as antecedents of job insecurity. Therefore, any generalization of the results to the Belgian population (or other countries) should be taken cautiously.

Second, due to the subjective nature of the constructs, the data were collected with a self-report questionnaire. This could increase the risk of common method bias and response bias, such as social desirability. Following the suggestions by Podsakoff et al. (2012) [79] to minimize socially desirable response bias, anonymity and confidentiality of the participation were emphasized prior to the participation in the survey. Furthermore, we implemented time lags as an objective separation between the predictor and criterion variables, which controlled for common method bias [79].

Third, the study implemented a six-month time lag between each measurement wave. To date, the literature on job insecurity has not specified the time frame over which our variables may influence each other. As mentioned earlier, this time lag might not have been optimal to observe the associations between the variables in the model over time. Quantitative job insecurity, due to the severity of the threat might affect the negative outcomes quicker than qualitative job insecurity. Similarly, a bidirectional relationship between quantitative and qualitative job insecurity cannot be ruled out unless properly analyzed with a diversity of time lags between the various measurement waves [80]. To our knowledge, this is the first study to answer the call for a longitudinal analysis of the relationship between quantitative and qualitative job insecurity and their concurrent effects on the wide range of outcomes. Future research may want to apply diversified time lags in order to establish the optimal time frame to examine the relationships between the variables of interest over time.

Furthermore, future research could add to the current literature by exploring the relationship between quantitative and qualitative job insecurity at the within-person level. Because job insecurity is a psychological construct, research on the relation between its dimensions should ideally address two components of this dynamic relation: between-person and within-person effects. In the current study, we solely applied a variable-centered approach and focused on the overall lagged associations between the levels of quantitative and qualitative job insecurity, job strains, and coping reactions (between-person effects). Hence, we estimated the relationship between the two dimensions of job insecurity to be the same for everyone in the sample, while it is expected that individuals will differ based on the underlying initial levels and trajectory of change of both, quantitative and qualitative job insecurity. Indeed, recent studies have identified up to five job insecurity profiles that vary in terms of how insecure employees feel and which type (quantitative vs. qualitative job insecurity) is dominant [81]. Subsequent research could re-examine the time-specific relationship between quantitative and qualitative job insecurity while simultaneously account for these individual differences.

## 5. Conclusions

The results of the current study highlighted the relevance of qualitative job insecurity, not only as an important work stressor but also as an antecedent of quantitative job insecurity. When analyzed together, only qualitative job insecurity predicted an increase in job strains and withdrawal coping reactions. In contrast to previous claims, the impact of qualitative job insecurity on employees and organizations seems more severe when compared with quantitative job insecurity. Our results also showed that, over time, qualitative job insecurity is associated with an increase in quantitative job insecurity, which should be considered when planning interventions at the organizational level.

## Figures and Tables

**Figure 1 ijerph-18-06392-f001:**
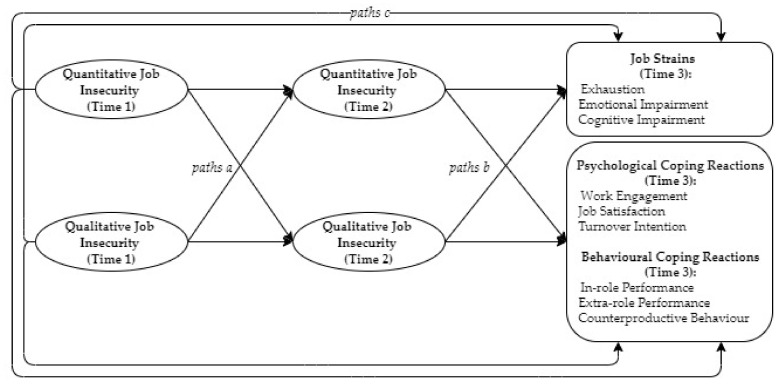
Representation of the theoretical model. Note: Paths *a*, *b*, and *c* represent the causal effects implied by the mediation processes. Indirect effects equal *a × b*; total effects equal *c + (a × b*).

**Figure 2 ijerph-18-06392-f002:**
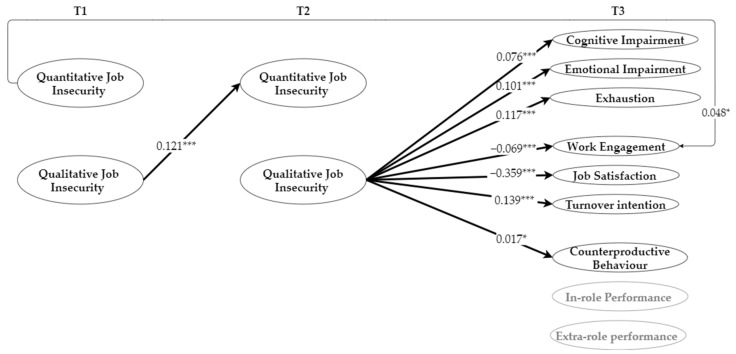
Autoregressive cross-lagged panel model with unstandardized path coefficients. Note: *p* < 0.05 *, *p* < 0.01 **, *p* < 0.001 ***; T1/T2/T3 indicate measurement waves; control variables as well as autoregressive and insignificant pathways are omitted for clarity.

**Table 1 ijerph-18-06392-t001:** Means, standard deviations, reliabilities (Cronbach’s alpha in parentheses), and correlation.

	**M**	**SD**	**Quan.1**	**Quan.2**	**Quan.3**	**Qual.1**	**Qual.2**	**Qual.3**	**EX.1**	**EX.2**	**EX.3**	**CC.1**	**CC.2**	**CC.3**	**EC.1**	**EC.2**	**EC.3**	**TI.1**	**TI.2**
Quan.1	2.45	1.01	(0.93)																
Quan.2	2.38	1.01	**0.68**	(0.94)															
Quan.3	2.37	0.99	**0.63**	**0.69**	(0.93)														
Qual.1	3.17	0.95	**0.51**	**0.37**	**0.37**	(0.9)													
Qual.2	3.08	0.97	**0.34**	**0.54**	**0.41**	**0.63**	(0.92)												
Qual.3	3.13	0.94	**0.30**	**0.38**	**0.52**	**0.54**	**0.63**	(0.91)											
EX.1	2.97	0.94	**0.25**	**0.17**	**0.16**	**0.45**	**0.37**	**0.31**	(0.9)										
EX.2	2.83	0.9	**0.21**	**0.25**	**0.17**	**0.40**	**0.44**	**0.36**	**0.74**	(0.9)									
EX.3	2.84	0.89	**0.17**	**0.19**	**0.23**	**0.33**	**0.37**	**0.42**	**0.66**	**0.73**	(0.89)								
CC.1	2.46	0.81	**0.18**	**0.15**	**0.13**	**0.31**	**0.25**	**0.20**	**0.45**	**0.36**	**0.31**	(0.91)							
CC.2	2.32	0.74	**0.13**	**0.21**	**0.16**	**0.24**	**0.31**	**0.22**	**0.4**	**0.48**	**0.36**	**0.64**	(0.9)						
CC.3	2.35	0.74	**0.09**	**0.14**	**0.18**	**0.21**	**0.24**	**0.29**	**0.34**	**0.37**	**0.43**	**0.62**	**0.70**	(0.91)					
EC.1	1.94	0.82	**0.27**	**0.22**	**0.19**	**0.37**	**0.28**	**0.24**	**0.51**	**0.4**	**0.38**	**0.49**	**0.37**	**0.34**	(0.88)				
EC.2	1.84	0.76	**0.22**	**0.27**	**0.19**	**0.32**	**0.36**	**0.25**	**0.41**	**0.51**	**0.41**	**0.35**	**0.45**	**0.36**	**0.62**	(0.89)			
EC.3	1.85	0.75	**0.20**	**0.23**	**0.26**	**0.31**	**0.28**	**0.35**	**0.35**	**0.39**	**0.49**	**0.30**	**0.34**	**0.43**	**0.56**	**0.65**	(0.87)		
TI.1	2.81	1.27	**0.36**	**0.22**	**0.23**	**0.42**	**0.24**	**0.19**	**0.37**	**0.27**	**0.24**	**0.36**	**0.22**	**0.20**	**0.35**	**0.23**	**0.20**	na	
TI.2	2.6	1.17	**0.21**	**0.33**	**0.26**	**0.27**	**0.38**	**0.35**	**0.26**	**0.31**	**0.28**	**0.25**	**0.30**	**0.29**	**0.25**	**0.34**	**0.26**	**0.60**	na
TI.3	2.64	1.18	**0.21**	**0.24**	**0.36**	**0.25**	**0.29**	**0.41**	**0.23**	**0.24**	**0.31**	**0.20**	**0.21**	**0.28**	**0.22**	**0.19**	**0.32**	**0.51**	**0.66**
JS.1	5.49	2.51	**−0.35**	**−0.24**	**−0.25**	**−0.53**	**−0.42**	**−0.34**	**−0.54**	**−0.45**	**−0.36**	**−0.49**	**−0.38**	**−0.32**	**−0.5**	**−0.38**	**−0.29**	**−0.65**	**−0.44**
JS.2	6.03	2.3	**−0.26**	**−0.36**	**−0.26**	**−0.45**	**−0.56**	**−0.47**	**−0.42**	**−0.51**	**−0.44**	**−0.39**	**−0.45**	**−0.39**	**−0.39**	**−0.49**	**−0.37**	**−0.44**	**−0.59**
JS.3	5.99	2.23	**−0.19**	**−0.26**	**−0.35**	**−0.39**	**−0.46**	**−0.55**	**−0.38**	**−0.42**	**−0.48**	**−0.36**	**−0.38**	**−0.42**	**−0.35**	**−0.35**	**−0.44**	**−0.38**	**−0.47**
WE.1	3.1	0.9	**−0.26**	**−0.19**	**−0.18**	**−0.42**	**−0.35**	**−0.28**	**−0.44**	**−0.34**	**−0.29**	**−0.57**	**−0.45**	**−0.41**	**−0.43**	**−0.33**	**−0.26**	**−0.50**	**−0.36**
WE.2	3.26	0.83	**−0.17**	**−0.23**	**−0.20**	**−0.35**	**−0.42**	**−0.35**	**−0.36**	**−0.39**	**−0.30**	**−0.46**	**−0.5**	**−0.46**	**−0.36**	**−0.37**	**−0.28**	**−0.37**	**−0.44**
WE.3	3.25	0.83	**−0.11**	**−0.19**	**−0.24**	**−0.31**	**−0.37**	**−0.43**	**−0.32**	**−0.32**	**−0.41**	**−0.44**	**−0.44**	**−0.53**	**−0.3**	**−0.28**	**−0.36**	**−0.31**	**−0.37**
IP.1	4.06	0.63	**−0.21**	**−0.11**	**−0.16**	**−0.14**	**−0.08**	**−0.11**	**−0.15**	**−0.14**	**−0.14**	**−0.25**	**−0.2**	**−0.22**	**−0.22**	**−0.21**	**−0.19**	**−0.06**	0.01
IP.2	4.11	0.6	**−0.12**	**−0.14**	**−0.15**	**−0.10**	**−0.12**	−0.07	**−0.12**	**−0.16**	**−0.16**	**−0.17**	**−0.23**	**−0.19**	**−0.17**	**−0.23**	**−0.19**	−0.02	−0.05
IP.3	4.08	0.59	**−0.16**	**−0.13**	**−0.22**	**−0.12**	**−0.10**	**−0.10**	**−0.11**	**−0.16**	**−0.18**	**−0.19**	**−0.2**	**−0.26**	**−0.14**	**−0.19**	**−0.23**	−0.03	−0.02
EP.1	3.74	0.76	**−0.09**	**−0.07**	−0.04	**−0.07**	**−0.07**	−0.05	**−0.08**	−0.05	−0.05	**−0.19**	**−0.16**	**−0.18**	**−0.08**	**−0.12**	**−0.09**	**−0.08**	0.01
EP.2	3.78	0.75	−0.03	−0.04	−0.03	−0.04	**−0.06**	**−0.09**	−0.03	−0.04	−0.06	**−0.11**	**−0.12**	**−0.15**	**−0.04**	**−0.1**	**−0.11**	−0.02	−0.03
EP.3	3.75	0.73	−0.05	**−0.07**	−0.05	−0.06	−0.06	**−0.05**	−0.05	**−0.07**	**−0.07**	**−0.16**	**−0.16**	**−0.19**	**−0.07**	**−0.14**	**−0.12**	**−0.07**	−0.07
CP.1	1.8	0.67	**0.05**	0.03	0.02	**0.15**	**0.13**	**0.08**	**0.14**	**0.08**	0.05	**0.47**	**0.34**	**0.36**	**0.27**	**0.20**	**0.17**	**0.24**	**0.16**
CP.2	1.69	0.59	0.04	**0.09**	**0.09**	**0.13**	**0.19**	**0.13**	**0.15**	**0.12**	**0.08**	**0.36**	**0.41**	**0.39**	**0.22**	**0.23**	**0.22**	**0.15**	**0.20**
CP.3	1.7	0.6	0	0.03	0.04	**0.12**	**0.15**	**0.17**	**0.12**	**0.10**	**0.11**	**0.36**	**0.38**	**0.43**	**0.18**	**0.19**	**0.23**	**0.11**	**0.17**
Gender	1.58	0.49	0	0.04	−0.01	**0.05**	**0.07**	**0.08**	**0.06**	**0.07**	0.04	0.03	**0.08**	0.03	**0.14**	**0.14**	**0.14**	**0.05**	0.04
Education	4.05	0.88	**−0.08**	−0.03	−0.06	**−0.05**	0.01	−0.01	**−0.09**	−0.03	−0.05	**0.08**	**0.09**	**0.13**	**−0.05**	−0.02	−0.03	−0.02	0
Positional tenure	10.66	9.43	0.04	0.02	0.05	**0.12**	**0.12**	**0.13**	0.01	0	−0.04	**−0.12**	**−0.14**	**−0.17**	0.02	**0.06**	0.02	**−0.13**	**−0.14**
	**TI.3**	**JS.1**	**JS.2**	**JS.3**	**WE.1**	**WE.2**	**WE.3**	**IP.1**	**IP.2**	**IP.3**	**EP.1**	**EP.2**	**EP.3**	**CP.1**	**CP.2**	**CP.3**	**SEX**	**EDU**	**EXP**
Quan.1																			
Quan.2																			
Quan.3																			
Qual.1																			
Qual.2																			
Qual.3																			
EX.1																			
EX.2																			
EX.3																			
CC.1																			
CC.2																			
CC.3																			
EC.1																			
EC.2																			
EC.3																			
TI.1																			
TI.2																			
TI.3	na																		
JS.1	**−0.32**	na																	
JS.2	**−0.41**	**0.74**	na																
JS.3	**−0.57**	**0.64**	**0.79**	na															
WE.1	**−0.25**	**0.74**	**0.61**	**0.55**	(0.86)														
WE.2	**−0.31**	**0.64**	**0.71**	**0.65**	**0.79**	(0.84)													
WE.3	**−0.40**	**0.54**	**0.63**	**0.72**	**0.71**	**0.79**	(0.85)												
IP.1	0	**0.16**	**0.15**	**0.14**	**0.18**	**0.16**	**0.15**	(0.86)											
IP.2	0	**0.11**	**0.17**	**0.15**	**0.11**	**0.17**	**0.13**	**0.59**	(0.88)										
IP.3	−0.02	**0.09**	**0.15**	**0.15**	**0.12**	**0.14**	**0.17**	**0.56**	**0.63**	(0.85)									
EP.1	−0.01	**0.16**	**0.14**	**0.11**	**0.24**	**0.20**	**0.18**	**0.24**	**0.21**	**0.18**	(0.78)								
EP.2	−0.03	**0.11**	**0.14**	**0.12**	**0.20**	**0.22**	**0.17**	**0.20**	**0.28**	**0.26**	**0.60**	(0.77)							
EP.3	**−0.07**	**0.13**	**0.16**	**0.17**	**0.22**	**0.22**	**0.24**	**0.15**	**0.20**	**0.20**	**0.58**	**0.66**	(0.75)						
CP.1	**0.13**	**−0.31**	**−0.23**	**−0.23**	**−0.41**	**−0.32**	**−0.32**	**−0.13**	**−0.08**	**−0.07**	**−0.16**	**−0.09**	**−0.16**	(0.66)					
CP.2	**0.14**	**−0.25**	**−0.27**	**−0.29**	**−0.33**	**−0.37**	**−0.37**	**−0.08**	**−0.12**	**−0.09**	**−0.13**	**−0.11**	**−0.16**	**0.68**	(0.63)				
CP.3	**0.18**	**−0.19**	**−0.26**	**−0.30**	**−0.31**	**−0.33**	**−0.38**	−0.05	−0.06	**−0.09**	**−0.14**	**−0.11**	**−0.15**	**0.66**	**0.71**	(0.64)			
Gender	0.04	−0.03	−0.03	−0.01	−0.04	−0.05	0	0.04	−0.02	0.03	**0.07**	0.05	**0.06**	**−0.06**	0	−0.02	na		
Education	0.04	**0.08**	−0.01	0	0.04	0.02	−0.01	−0.01	−0.05	−0.04	**−0.04**	−0.03	−0.03	0.04	**0.07**	**0.09**	0.03	na	
Positional tenure	**−0.12**	−0.01	**−0.08**	−0.05	**0.07**	0.02	0	0.01	0	0.03	−0.01	−0.01	−0.01	**−0.11**	**−0.09**	**−0.07**	**−0.11**	**−0.19**	na

Note: N = 2003. Bold numbers indicate statistically significant correlation at the 5% level. QN = quantitative job insecurity; QL = qualitative job insecurity; EX = exhaustion; CC = cognitive impairment; EC = emotional impairment; TI = turnover intention; JS = job satisfaction; WE = work engagement; IP = in-role performance; EP = extra-role performance; CP = counterproductive behavior.

**Table 2 ijerph-18-06392-t002:** Fit indices of competing nested factor models and standardized maximum likelihood estimates.

**Factorial Structure of the Measurement Model**													
**Model No.**	**Model**	***χ*^2^**	**df**	**CFI**	**TLI**	**RMSEA**	**SRMR**	**Comparison to Model No.**	**Δ*χ*^2^**	**Δdf**	***P***	**ΔCFI**	**ΔRMSEA**
M1	Hypothesized: 33-factor model	7246.195	3852	0.967	0.961	0.021	0.043						
M2	Alternative: 15-factor model	24,273.882	4269	0.805	0.792	0.048	0.094	M1	17,027.687 ***	417	<0.001	0.162	0.027
M3	Alternative: 12-factor model	31,563.306	4308	0.734	0.719	0.056	0.109	M1	24,317.111 ***	456	<0.001	0.233	0.035
M4	Alternative: 3-factor model	49,055.402	4371	0.564	0.545	0.071	0.112	M1	41,809.207 ***	519	<0.001	0.403	0.05
Longitudinal Measurement Invariance of the Hypothesized 33-factor Model													
M5	Metric invariance	7305.662	3894	0.967	0.961	0.021	0.043	M1	59.467 *	42	0.039	0	0
M6	Strong invariance	7481.112	3954	0.966	0.96	0.021	0.043	M5	175.450 ***	60	<0.0001	0.001	0
M7	Strict invariance	7838.118	4014	0.963	0.958	0.022	0.043	M6	357.006 ***	60	<0.0001	0.003	0.001

Note: N = 2003; *p* < 0.05 *, *p* < 0.01 **, *p* < 0.001 ***; *χ*^2^ = chi-square; df = degrees of freedom; CFI = comparative fit index; TLI = Tucker–Lewis index; RMSEA = root mean squared error of approximation; SRMR = standardized root mean squared residual.

**Table 3 ijerph-18-06392-t003:** Test of alternative structural and time invariance.

**Analysis of the Alternative Structural Models**													
**Model No.**	**Model**	***χ*^2^**	**df**	**CFI**	**TLI**	**RMSEA**	**SRMR**	**Comparison to Model No.**	**Δ*χ*^2^**	**Δdf**	***P***	**ΔCFI**	**ΔRMSEA**
M8	Autoregressive with covariates	11,797.757	4580	0.930	0.927	0.028	0.086						
M9	Chirumbolo’s Longitudinal JIIM	11,665.309	4551	0.931	0.928	0.028	0.076	M8	132.45 ***	29	<0.001	0.001	0
M10	Alternative mediation model	11,737.036	4551	0.930	0.927	0.028	0.078	M8	60.721 ***	29	<0.001	0	0
M11	Hypothesized: dual-mediation model	11,606.381	4522	0.931	0.928	0.028	0.072	M9	58.928 ***	29	<0.001	0	0
Stability of the Hypothesized Dual-Mediation Model													
M12	M11 + equal autoregressive paths	11,630.851	4533	0.931	0.928	0.028	0.072	M11	24.47 *	11	0.011	0.001	0
M13	M12 + equal paths “a”	11,630.988	4535	0.931	0.928	0.028	0.072	M12	0.14	2	0.934	0	0
M14	M13 + equal paths “b”	11,663.687	4553	0.931	0.928	0.028	0.072	M13	32.7 *	18	0.018	0	0

Note: N = 2003; *p* < 0.05 *, *p* < 0.01 **, *p* < 0.001 ***; *χ*^2^ = chi-square; df = degrees of freedom; CFI = comparative fit index; TLI = Tucker–Lewis index; RMSEA = root mean squared error of approximation; SRMR = standardized root mean squared residual.

## Data Availability

Restrictions apply to the availability of these data. Data were obtained from Anahí Van Hootegem, Steffie Desart and Hans De Witte from KU Leuven and are available from the authors upon request.

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
