# Peer review of "On the Reciprocal Relationship between Quantitative and Qualitative Job Insecurity and Outcomes. Testing a Cross-Lagged Longitudinal Mediation Model"

_ijerph, 2021, doi:10.3390/ijerph18126392_

Round 1

Reviewer 1 Report

Dear Authors,

this paper is well written and statistically proven. Yet, the subject of this article is an "old wine in a new bottle".....There is always a kind of impact (positive, negative or neutral; lines 35-36 (...) "These changes directly impact employees’ work and the context in which their job is performed" - this statement it is a truism).

Line 295: Data for the present longitudinal study were collected from Flemish employees (the Dutch-speaking region of Belgium).

What is "special" about the abovementioned region? The comparison to other regions should be useful.

Moreover, what were the industrial sectors (lines 337-348) and kinds of jobs (f.e. "I feel insecure about the future of my job" - and this job is? what is the reward (salary, commission)). It is particularly important referring to the demand/supply issues on the work market....The only differentiation public/private can be confusing....

lines 762-778

"Specifically, we found that qualitative job insecurity may generate negative outcomes for both individuals (e.g., burnout syndrome, job dissatisfaction) and organizations (e.g., turnover intentions, work disengagement, counterproductive behaviors)". This confusion (lack of source) has been described already

Look at:

DOI: 10.1111/1464-0597.0077z

DOI: 10.7334/psicothema2020.205.

DOI: 10.1080/00223980.2020.1774484

 and over 200 more...

Author Response

Dear Authors,

this paper is well written and statistically proven. Yet, the subject of this article is an "old wine in a new bottle"....

  1. The current study examines the job insecurity-negative outcomes relationship, which indeed might be considered a well-researched topic. At the same time our study looks at that topic from a perspective never taken before. First, we look at the simultaneous effect of quantitative and qualitative job insecurity on the negative outcomes. Both dimensions are established as important work stressors that measure different aspects of job insecurity. However, this is the first study to test the quantitative/qualitative job insecurity à outcomes relationship while controlling for the effects of the other dimension, which not only sheds new light on the unique contribution of each dimension of job insecurity but also compares the strength of those effects. Second, we examine the relationship between quantitative and qualitative job insecurity. Previous research proposed a unidirectional relationship, in which quantitative job insecurity precedes and leads to qualitative job insecurity. Our study is the first to expand on that model and in line with conservation of resources theory proposes an equally plausible reverse relationship. Lastly, we use longitudinal data, collected at three points with a six months’ time lag in between. A longitudinal design allows us to control for the previous levels of the outcome variables while simultaneously testing for the effects of quantitative and qualitative job insecurity, which gives more robust estimations of the effects of the two dimensions of job insecurity. Furthermore, with longitudinal data we can estimate the temporal order of the quantitative job insecurity-qualitative job insecurity relationship and examine whether: 1) quantitative job insecurity precedes and leads to qualitative job insecurity (as argued in previous research); 2) there is a reverse relationship, meaning that qualitative job insecurity precedes and leads to quantitative job insecurity; or, as we argue in this study, 3) there is a reciprocal relationship between quantitative and qualitative job insecurity.

….There is always a kind of impact (positive, negative or neutral; lines 35-36 (...) "These changes directly impact employees’ work and the context in which their job is performed" - this statement it is a truism).

  1. Thank you for that comment. Indeed, the sentence needs clarification. Therefore, I altered the text and included some references:

Text change: These changes negatively impact employees’ work and the context in which their job is performed. References:

DOI: 10.1108/00483481111133318

DOI: 10.1108/02683940010320589

DOI: 10.5465/amd.2017.0034

Line 295: Data for the present longitudinal study were collected from Flemish employees (the Dutch-speaking region of Belgium).

What is "special" about the abovementioned region? The comparison to other regions should be useful.

  1. I agree that the comparison with other regions would be beneficial. In fact, in the discussion part, I do imply that cultural differences that might be observed between the regions within Belgium could moderate the effect of quantitative and qualitative job insecurity on the outcomes. However, due to lack of data from the other regions we were unable to test for that (the data were only collected in the Flemish region). Furthermore, that type of research would require equivalence testing of all measures, which would add additional complexity to this already complex analysis, and thus goes beyond the scope of this paper. Note that previous research on job insecurity has been successfully performed using data from the Flemish region, receiving meaningful results that lead to relevant conclusions regarding the development and impact of job insecurity.

https://doi-org.kuleuven.e-bronnen.be/10.1080/02678373.2012.703900

https://doi.org/10.1108/CDI-03-2015-0046

https://doi-org.kuleuven.e-bronnen.be/10.1002/smi.2584

https://doi-org.kuleuven.e-bronnen.be/10.1002/smi.1371

10.1017/sjp.2019.7

10.1080/1359432X.2016.1143815

10.3390/ijerph16152640

Moreover, what were the industrial sectors (lines 337-348) and kinds of jobs (f.e. "I feel insecure about the future of my job" - and this job is? what is the reward (salary, commission)). It is particularly important referring to the demand/supply issues on the work market....The only differentiation public/private can be confusing....

  1. Thank you for that remark. Following your feedback, I included information regarding the industrial sector (four-sector model) and job type. The sample is representative of the Flemish working population regarding the distributions of employees across the four sectors. We also observed that the majority of our sample are white collar workers. Unfortunately, we had no data regarding salaries or other work benefits that could have had impact on the modelled variables.

Text change: Respectively, 6% were blue-collar workers (2% unskilled workers; 4% skilled workers), 61% were white-collar workers (26% lower level and administrative clerk; 35% middle level employee) and 33% were in managerial positions (24% low and middle level management; 9% senior management). Below 1% of the respondents worked in the primary sector (extraction of raw materials/ farming/fishing), 28% worked in the secondary sector (‘industry’: manufacturing/utilities), 35% worked in the tertiary sector (‘services’: retail/financial services/communication/hospitality/real estate/information technology) and 28% worked in the public sector (‘government’: education/public administration/research and development).

lines 762-778 "Specifically, we found that qualitative job insecurity may generate negative outcomes for both individuals (e.g., burnout syndrome, job dissatisfaction) and organizations (e.g., turnover intentions, work disengagement, counterproductive behaviors)". This confusion (lack of source) has been described already

Look at:

DOI: 10.1111/1464-0597.0077z

DOI: 10.7334/psicothema2020.205.

DOI: 10.1080/00223980.2020.1774484

 and over 200 more...

  1. Thank you for this useful comment. Following your suggestions, we rephrased the paragraph and added references:

Text change: This study particularly emphasizes the importance of qualitative job insecurity. Threats to job characteristics not only lead to negative consequences for employees’ health, work attitudes and performance, but also engender the threat of job loss, six months later. A focal point of HRM practices should be the implementation of new practices that primary aim at reducing the experience of qualitative job insecurity among the employees and buffer its effect on the outcomes. This could be achieved by increasing investments in career development, which not only elicit the engagement in the organization but also send a clear signal on a secured future role in the organization. Furthermore, organizations should establish clear formal communication channels to address prospective workplace changes, which have been shown to reduce employees’ feelings of job insecurity [78]. Finally, negative outcomes of qualitative job insecurity might be reduced if organizations create opportunities for employees to participate in a decision making processes regarding workplace changes that directly affect the future characteristics of their job [79]. The antecedental role of qualitative job insecurity in the development of quantitative job insecurity suggests that these practices might also indirectly decrease the experience of quantitative job insecurity and may thus hamper its negative effects.

Reviewer 2 Report

This paper simultaneously examines the longitudinal effects of quantitative and qualitative job insecurity, including a wide range of the outcome variables, classified as job strains, and psychological and behavioural coping reactions. Data come from a three-wave panel design that surveyed 2003 Belgian employees. The Authors provide valuable information on whether the importance of a particular dimension of job insecurity is related to the specific outcome under consideration, assess the relationship between quantitative and qualitative job insecurity and address the limitations of the previous cross-sectional research by implementing a three-wave longitudinal research design.

The research question is innovative, clear, and well defined, the topic is absolutely original and the paper is accurate. Moreover, it contains a perfect literature review concerning the topic. The discussion of results is sound and well-conducted, and there is a clear description of the limitations and future research opportunities.

I think that this is one of the best papers I had the opportunity of reviewing, also considering that I got my PhD with a thesis related to this particular topic. I think that the paper will attract a wide readership.

The paper does read fluidly, and it is written in Standard English.

For all those reasons, I think there is an overall benefit to publishing this work.

  • Does the introduction provide sufficient background and include all relevant references?

The introduction clearly presents the research question, and all the paper is very good.

  • Is the research design appropriate?

The research on which the paper is based seems well designed. The analysis proposed follows well-known methods in this strand of research, and the paper proposes a new method.

  • Are the methods adequately described?

Methods are correctly described.

  • Are the results clearly presented?

Results are clearly presented.

  • Are the conclusions supported by the results?

The conclusions are drawn appropriately based on the data presented.

Review Comments to the Author

Dear Authors, I really appreciated reading your paper and I think it should be published, I am really impressed.

I have just a couple of remarks:

  • Pages 3-4, lines 142-161: I find this paragraph misleading. You write that you “do not formulate specific hypotheses”, but, you do. I think that you should control if this paragraph makes sense in the overall explanation;

  • After page 11, page numbers are broken;

  • I think that the footnote on page 7 should be rewritten using “The Authors would like to thank…”

  • Some references in the text do not have the number and I don’t’ find them in the reference section. (i.e at lines 144, 153, 156, 266, 679)

Author Response

Dear Reviewer,

Thank you for your positive remarks. We are happy to see that an expert in a field of job insecurity finds this study well-designed, relevant, clear, and easy to read. We appreciate your constructive feedback that helps us to improve the manuscript.

Pages 3-4, lines 142-161: I find this paragraph misleading. You write that you “do not formulate specific hypotheses”, but, you do. I think that you should control if this paragraph makes sense in the overall explanation;

  1. Thank you for this comment. We reviewed this paragraph and indeed the idea was not clearly presented. In the earlier paragraph and the one mentioned here we outline that both dimensions are important work stressors and note that separate analyses consistently link both with negative outcomes for both employees and organizations. Therefore, in the current study we expect that both, quantitative and qualitative job insecurity, would be negatively associated with the outcome variables. However, previous literature shows inconclusive results regarding the importance of quantitative and qualitative job insecurity and regarding the comparative strength of their effects. This holds us from formulating specific hypotheses regarding the strength and relative importance of those relations. We propose the following correction:

Text change: In line with current knowledge, we expect that both quantitative and qualitative job insecurity have a negative effect on the outcomes. However, no specific hypothesis regarding the comparative strength and importance of these relationships are formulated in the current study.

After page 11, page numbers are broken;

  1. Thank you for addressing that issue. We will send an email to the journal because it may be a problem with the editing. When we downloaded the manuscript, the page numbering was not broken. The two other reviewers have not addressed this issue which makes me think that it might have been a bug that hampered the downloading process.

I think that the footnote on page 7 should be rewritten using “The Authors would like to thank…”

  1. Thank you for that comment. We made the following correction:

Text change: The Authors would like to thank Steffie Desart and Anahí Van Hootegem from KU Leuven for sharing their dataset and providing us with all the information about the data collection process.

Some references in the text do not have the number and I don’t’ find them in the reference section. (i.e at lines 144, 153, 156, 266, 679)

  1. Thank you for that comment. We used Mendeley to insert citations and the journal template uses numbering. However, in the cases you refer to, we included the citation in the text; that is why it is not numbered in the text, but it had a number in the reference list. However, following your comment we include the reference number in the text.

Reviewer 3 Report

The work presented is correct, they should improve some aspects:

  1. The proposed model must be specified, because afterwards it is validated in very different ways.
  2. the final model does not look like the one proposed. They should improve the presentation of the same, and include all the data of values between variables.
  3. They should indicate the reliability and validity values of the different instruments used, but for this study.
  4. They must include the correlation variables in the model, there are variables that do not appear. They are already discarded initially in the theoretical model, if so, why are they included in the study? If not, the model should be based on them and take into account all the variables analyzed.

Without improving this does not make sense.

Author Response

The proposed model must be specified, because afterwards it is validated in very different ways.

1. In the current study we proposed a dual-mediation model that combines the simultaneous effects of quantitative and qualitative job insecurity on the negative outcomes with a reciprocal relationship between the two aspects of job insecurity (see Figure 1). We proceed with the analysis using a cross-lagged panel method, which is part of the structural equation modelling approach. We followed a stepwise procedure, recommended when the model is specified with latent variables (Cole and Maxwell, 2003; and Little et al., 2007). In the analysis section of the article, we describe all the steps that precede the specification of the proposed model. Moreover, three alternative structural models are estimated in order to test whether the proposed model fits the data best or whether models with a specified unidirectional relationship between quantitative and qualitative job insecurity (Chirumbolo’s JIIM and Alternative Model in which we test whether qualitative job insecurity leads to quantitative job insecurity, that later leads to negative outcomes) fits the data significantly better. Our analysis indicates that the proposed model indeed fits the data better than the alternative models, which allows us to proceed with the hypothesis testing using the proposed model. We acknowledge that it is a complex procedure, hence we added the following clarification:

Text change:

Subsequently, we estimated and compared the model fit of four structural models, in order to select the best model to test the hypothesized mediation effects. In this step, we added positional tenure as a control variable to each of the estimated models. Firstly, we estimated a structural model with autoregressive paths (M8). Building on that model, we then estimated the longitudinally extended Chirumbolo et al.’s JII model with qualitative job insecurity as mediator (M9). Then, we analyzed the reversed mediation model with the mediating role of quantitative job insecurity (M10). Lastly, we fitted the hypothesized dual mediation model (M11) that integrates the reciprocal relationships between quantitative and qualitative job insecurity. This procedure allows to test whether the model proposed in the current study fits the data better than the alternative models with an estimated unidirectional relationship between quantitative and qualitative job insecurity.

the final model does not look like the one proposed. They should improve the presentation of the same and include all the data of values between variables.

2. The final model that has been used for hypothesis testing is the proposed theoretical research model. However, the graph that represents the estimated paths (Figure 2) includes only significant associations. The non-significant associations, autoregressive coefficients, and a control variable (tenure) were not presented in order to ease the interpretation. This practice has been successfully implemented in other studies: 

10.3390/ijerph16101842

10.1016/j.jbusres.2020.12.045

A graphical representation of the final model (with all estimated path coefficients) is very complex and hard to read. Nevertheless, we do acknowledge the value in presenting all results. Hence, we propose to keep the figure with only significant path coefficients in the text and additionally include a complete figure in an Appendix.

They should indicate the reliability and validity values of the different instruments used, but for this study.

3. Thank you for this comment. Internal validity and reliability of the scales has been examined with two methods. For multi-item scales the internal validity was examined with Cronbach’s alpha, whereas single-item scale’s reliability (job satisfaction and turnover intention) was measured with the test-retest method. We corrected the text to include these values in the text.                                   

They must include the correlation variables in the model, there are variables that do not appear. They are already discarded initially in the theoretical model, if so, why are they included in the study? If not, the model should be based on them and take into account all the variables analysed.

4. Thank you for that comment. The inclusion of control variables had been done following the recommendations of Bernerth el at (2016). We begin with a literature review and check for the variables that had been reported to associate with the variables modelled in the current study. We search for the theoretical rationale for the inclusion of these variables and the results of previous research. Based on this procedure we decided to include three control variables: gender, education and tenure, because they have been linked in previous studies with the levels of job insecurity and outcome variables like job satisfaction, turnover, or job performance. Next, we estimated bivariate correlations. We found that only positional tenure correlates significantly with the predictor and criterion variables. Following Berneth et al’s (2016) suggestion, we therefore excluded gender and education. Following your comment, we rephrased the section on control variables which hopefully clarifies the interpretation of the process we followed when including control variables.

Text change:

We considered three potentially relevant control variables, including gender (0=male; 1=female), positional tenure (years) and education level (1=primary education; 2=lower secondary education; 3=higher secondary education; 4=non-university higher education; 5=university higher education; 6=doctorate), treated as a continuous variable that represents a range from less educated to highly educated. First, according to job dependence theory male employees experience higher economic insecurity [5], which translates into higher perceived job insecurity, as they feel more responsible to provide financial stability for the family [11,48]. Second, human capital theory explains that more educated employees with longer tenure exhibit more positive work attitudes and behaviors. Empirical evidence shows that higher education and longer tenure grants access to better jobs with higher salaries and additional resources, which results in higher job satisfaction and task performance, and gives more incentives to remain in an organization [49–51].

Despite this theoretical rationale and the evidence from previous research, the examination of the bivariate correlations (see Table 1) shows no significant associations between gender and the two dimensions of job insecurity. Furthermore, we found no significant correlation between education and job satisfaction or performance. On the other hand, consistent with our theory-based expectations, positional tenure was negatively correlated with turnover intentions. Hence, to facilitate the interpretation of the results and to maximize statistical power we performed the analysis without controlling for gender and education. However, given the significant bivariate corrections and our theoretical rationale, we controlled for positional tenure.

Round 2

Reviewer 3 Report

 Accept in present form